# Exploiting Signal Propagation Delays to Match Task Memory Requirements in Reservoir Computing

**DOI:** 10.3390/biomimetics9060355

**Published:** 2024-06-14

**Authors:** Stefan Iacob, Joni Dambre

**Affiliations:** IDLab-AIRO, Ghent University, 9052 Ghent, Belgium; joni.dambre@ugent.be

**Keywords:** distance-based delays, inter-neuron delays, echo state networks, recurrent neural networks, reservoir computing, memory capacity

## Abstract

Recurrent neural networks (RNNs) transmit information over time through recurrent connections. In contrast, biological neural networks use many other temporal processing mechanisms. One of these mechanisms is the inter-neuron delays caused by varying axon properties. Recently, this feature was implemented in echo state networks (ESNs), a type of RNN, by assigning spatial locations to neurons and introducing distance-dependent inter-neuron delays. These delays were shown to significantly improve ESN task performance. However, thus far, it is still unclear why distance-based delay networks (DDNs) perform better than ESNs. In this paper, we show that by optimizing inter-node delays, the memory capacity of the network matches the memory requirements of the task. As such, networks concentrate their memory capabilities to the points in the past which contain the most information for the task at hand. Moreover, we show that DDNs have a greater total linear memory capacity, with the same amount of non-linear processing power.

## 1. Introduction

Many real-world machine learning problems are temporal in nature and require learning spatio-temporal patterns. Recurrent neural networks (RNNs) are commonly used to model temporal tasks and rely on recurrent connections to represent temporal dependencies in the data, passing information from one time step to another. In contrast, an important property of temporal processing in biological networks, besides recurrent connections, are inter-neuron delays, which have mostly been overlooked in artificial RNNs. It has been shown that the brain relies on axon delay in neural circuits for coincidence detection [1], for sensory processing [2], and for shaping the spatio-temporal dynamics of motor control [3]. Most mammals show a large variety in inter-neuron signal propagation delays [4]. Many of these delays are caused by the integration of signals such as in dendrites, somata, and synapses, which additionally perform non-linear transformations. However, axons propagate action potentials unchanged and only introduce a delay that varies with the axon length, diameter, and myelination. In this paper, we present an analysis of the effect of “axon-like” inter-neuron delays on temporal processing of RNNs and show that delays can be adapted to task memory requirements.

One of the RNN-based strategies for modeling temporal data is reservoir computing (RC), a paradigm originally proposed as an alternative to gradient-based RNN training such as back-propagation through time (BPTT) [5]. RC makes use of a dynamical system called a reservoir, and a readout layer that can be trained to estimate task labels based on reservoir dynamics. Software simulated RC systems are generally based on fixed RNN reservoirs, with weights that are randomly sampled from a task-optimized distribution. The reservoir performs non-linear, history-dependent transformations on the input sequence. This can also be viewed as a temporal kernel [6], which projects the data to a higher dimensional space, increasing the linear separability of the task. In principle, any model can be used as a readout layer, but generally linear models such as ridge regression are used [7]. This projection to a higher dimensional space can in fact be carried out with any type of dynamical system, not exclusively RNNs. This is the basis of physical reservoir computing, where instead of simulating an RNN, the reservoir is a physical substrate (e.g., an electronic [8], photonic [9], or memristive [10] circuit) [11]. Physical RC has the benefit of being fast and energy efficient, due to not having to simulate a reservoir, and physical reservoirs generally being very low power (and sometimes even fully passive). Although software simulated RC has somewhat decreased in popularity, recent years have seen ongoing development in physical RC. Nevertheless, software simulated RC serves as a model system for physical systems and can be used as a convenient testbed for novel RC features.

Echo state networks (ESNs) [12] represent one of the most commonly used types of reservoirs. These RNNs make use of simple sigmoid or hyperbolic tangent activation functions as opposed to the spiking neuron models used in liquid state machines [13], another major class of reservoirs. However, from the perspective of biological plausibility, ESNs can also be framed as a model for rate-based neural networks. Besides the recurrent connections used in conventional RNNs, ESNs usually also employ leaky nodes, which act as low-pass filters, internally integrating information over time. This offers another way to optimize the temporal dynamics of the reservoir. In fact, this leak mechanism serves as a way of separating the effect of self-connections from recurrent connections, by providing a separate tuneable parameter. Moreover, it was shown that a distribution of different leak rates in a network further improves performance on temporal tasks by improving temporal richness of the reservoir [14]. The importance of temporal diversity of networks is not limited to RC. Similarly, the use of adaptive and diverse temporal parameters also improved performance and memory capacity of conventional RNNs [15]. Although both recurrent connections and leaky neurons pass over information to the next time point, none of these mechanisms can do so over multiple time steps without transforming the information through non-linear activation functions. Past work in the field of RC has shown the importance of obtaining multi-scale dynamics through delays, to the extent that single-node reservoirs were able to achieve a similar performance as multi-node reservoirs when time-delayed feedback was used [16,17]. Just by tuning the feedback strength in this type of single-node delayed systems, a range of different behaviors can be achieved. These findings are useful for physical RC implementations, as they are easier to implement in hardware than RNN reservoirs. In fact, delays are a fundamental property of any physical and biological system. Whereas before, delays in physical reservoirs were seen as a limitation, since the introduction of time delay reservoirs in [16], it has become clear that the intrinsic delay properties of physical systems can be exploited to improve reservoir performance.

Besides feedback delay, a time-delayed input was shown to reduce hyperparameter optimization requirements of reservoirs [18,19]. Furthermore, the inclusion of past inputs into the reservoir allows for smaller reservoirs without reducing performance [20]. Our recent work related to time-delays has even further explored the importance of delayed reservoirs by introducing delays in multi-node systems (ESNs) as inter-node delays instead of only feedback delays, further highlighting the importance of multiscale dynamics [21,22]. It was shown that in these distance-based delay networks (DDNs), the use of variable inter-node delays can improve task performance on temporally complex tasks compared to conventional, single-step delay systems [21].

Although this consistent increase in ESN task performance was shown empirically, the question remains what the underlying mechanism is that leads to this improvement. Classical RNNs, which are used as reservoirs for ESN, pass the network activity through non-linear activation functions at every timepoint, thus continuously decreasing the amount of information about the past states. This means that the memory capacity of RNNs will monotonically decrease in time. Many real-world machine-learning problems are temporally complex and require memory of previous inputs and system states at specific points in the past. Others require the synchronization of inputs at specific temporal intervals (e.g., coincidence detection [1]). Due to their monotonically decreasing memory capacity, in RNNs, we can only increase the memory capacity at time t by increasing t-1 as well. We hypothesize that the improvement in task performance seen in DDNs can be explained by the effect that the delays have on memory capacity. The optimization of delayed connections can result in non-monotonically decreasing memory capacity profiles, which allows for peaks of memory capacity at specific points in time. Following this reasoning, we would expect optimized DDN to have peaks in memory capacity coinciding with high memory demand of the task for which they are optimized.

Linear memory capacity is a metric introduced by [23], often used to characterize computational properties of reservoirs (e.g., [24,25]). Memory capacity quantifies how well past inputs to the reservoir can be reconstructed based on current reservoir activity. This concept was later extended by [26] to non-linear computations. In this paper, we compare the linear memory capacity with the task memory requirements, which we here call *task capacity*. The novel contribution of this paper is an in-depth analysis on the effect of variable inter-node delays (both optimized and random) on linear memory capacity, and how this relates to the task capacity. We show that evolving inter-neuron delays in DDNs indeed results in memory capacity profiles that match the task profile for which they are optimized. More specifically, without inter-node delays, it is not possible to achieve peaks of memory capacity at specific lags. To our knowledge, the ability to distribute memory to specific time points, without relying on simply increasing the total memory capacity, and instead using inter-node delays, has not been studied before. Considering the fact that DDNs serve as model systems for physical reservoirs, this has important implications for the development of novel time-delay reservoirs that include in-reservoir delays, in addition to previously explored input delays and feedback delays [16,17,18].

The remainder of this paper is structured as follows. In Section 2, we discuss the formalization for DDNs (Section 2.1), Adaptive DDNs Section 2.2), the corresponding optimization procedure (Section 2.3), the benchmark tasks that we use (Section 2.4), and the concept of linear memory capacity (Section 2.5). Next, we present our experimental setup and results in Section 3. Lastly, we discuss our findings in Section 4.

## 2. Methods

In this section, we discuss the ESN models used in this paper, along with the training, hyperparameter optimization and benchmark tasks used. ESNs were first presented as a computationally efficient alternative to classical RNNs that are trained using BPTT. Instead, ESNs make use of a fixed, recurrent ‘reservoir’ of (usually leaky) sigmoid or hyperbolic tangent neurons. Only a linear readout layer is trained. The ESN update equation is as follows.
(1)$$\begin{array}{cc} x(n) & =\left(1-a\right)x\left(n-1\right)+a\cdot f(W_{\mathrm{res}}x\left(n-1\right)+b_{\mathrm{res}}+W_{\mathrm{in}}v\left(n-1\right)) \end{array}$$

Here, x(n), *a*, f(·), v(n) and $b_{\mathrm{res}}$ are, respectively, the reservoir activity at time *n*, the leak rate, the activation function, the task input at time *n*, and the reservoir bias weights. $W_{\mathrm{res}}$ is the *N* by *N* recurrent reservoir weight matrix, and $W_{\mathrm{in}}$ is the *N* by *T* input weight matrix, with *N* being the number of neurons and *T* the input dimensionality.

Although ESNs as such have lost popularity as an alternative to BPTT, the simplicity of training makes them particularly convenient as a model system for physical RC and as a test bed for novel features such as distance-based delays, because they do not require the development of a gradient-based optimization technique. Both DDNs and ADDNs are extensions of ESNs and similarly require task-specific hyperparameter optimization next to training a readout layer. Since a reservoir is randomly initialized, it can be seen as a sample from a distribution, defined by its hyperparameters (e.g., connectivity parameters, weight scaling, etc.). Although training a single readout layer is fast and requires few data points, optimizing hyperparameters requires training and evaluating task performance of readout layers for a number of reservoir samples from each candidate hyperparameter set. In this paper, we use the Covariance Matrix Adaptation Evolutionary Strategy (CMA-ES) [27] for optimizing reservoir hyperparameters. This evolutionary strategy works by sampling new candidate hyperparameter sets from a multivariate Gaussian. Each candidate is assigned a fitness score. The covariance matrix of this distribution is adapted in subsequent generations, such that the likelihood of previous successful candidates is increased.

In the following subsections, we describe the previously introduced DDNs and ADDNs, and we discuss the optimization procedure. Next, we describe the tasks used to evaluate these models. Lastly, we explain the Linear Memory Capacity metric.

### 2.1. Distance-Based Delay Networks

DDNs are based on conventional leaky ESNs, as defined by the state update equation from [28]. Note that the ESN contains information about previous states only through the leak mechanism and the recurrent reservoir connections. As mentioned, DDNs include the mechanism of variable inter-node delay according to the models presented in [21,22], which can combine the current reservoir state with node activity up to $D_{\max}$ time steps ago, with $D_{\max}$ being the largest delay present in the reservoir. DDNs are modeled in Euclidean space (2D in this paper). Each neuron is characterized by a position vector containing the x and y coordinate for each neuron *i*, described by $l_{i}=\left(x_{i},y_{i}\right)$. A distance matrix S is defined, with element $S_{i,j}=∥l_{i}-l_{j}∥$. An ESN can be defined with inter-neuron delays proportional to the inter-neuron distances. However, because this is simulated in discrete time, the distance matrix also needs to be discretized as such: $D=\left\lfloor \frac{S}{\Delta t\cdot v_{p}}\right\rfloor$, where Δt and $v_{p}$ are, respectively, the duration in seconds of one time step and the pre-defined signal propagation velocity between neurons in meters per second. $\left\lfloor \cdot \right\rfloor$ refers to the rounding down operator. Instead of a single reservoir weight matrix, a set of sparse, delay-specific weight matrices can be defined as follows.
(2)$$\begin{array}{cc} W_{i,j,D=d} & =\delta_{d,D_{i,j}}\cdot W_{i,j} \end{array}$$

Here, $\delta_{\cdot ,\cdot }$ is the Kronecker delta operator. As such, the elements of matrix $W_{D=d}$, for which the corresponding inter-neuron distance is *d* delay steps, are given by the original weight matrix. All other elements are set to 0. Based on these delay-specific weight matrices, the DDN reservoir activity is defined by the state update Equations (Equation 3) and (4) [21].
(3)$$\begin{array}{cc} x(n) & =(1-a)x(n-1)+a\sigma (y(n-1)) \end{array}$$
(4)$$\begin{array}{cc} y(n) & =\sum_{d=0}^{D_{\max}}\left(W_{D=d}^{\mathrm{res}}x\left(n-d\right)+W_{D=d}^{\mathrm{in}}v\left(n-d\right)\right)+b_{\mathrm{res}} \end{array}$$

Here, x(n), *a*, σ(·), v(n), and $b_{\mathrm{res}}$ have the same meanings as in Equation (Equation 1). y(n) can be seen as a pre-activation term at time *n*, i.e., the summed inputs for each neuron activation function. The software implementation of DDNs are provided in the Appendix A.

### 2.2. Adaptive Distance-Based Delay Networks

Originally, the main proposed benefit of ESNs has been the use of randomly initialized, fixed reservoirs. Avoiding the training of recurrent weights saves computation time and allows for physical implementations without the difficulties of modeling weight updates in hardware. However, in simulated ESNs, efforts have been made to depart from the idea of a fixed reservoir. Several RC implementations make use of Hebbian plasticity such that reservoir weights can adapt to better model input dynamics. In [29], the anti-Oja rule [30] was used in ESNs to improve Mackey–Glass performance in ESNs. In addition to Oja’s rule, Yusoff et al. also tested the BCM rule [31] in both temporal (Mackey–Glass and sun spot datasets) and classification (Breast Cancer and Adult Census Income datasets) tasks [32]. Both neural plasticity rules were tested in combination with on-line and off-line learning. BCM was found to outperform Oja, especially in off-line learning. The ESN implementation introduced in [33] combines Hebbian plasticity and intrinsic plasticity, which is shown to outperform conventional ESNs and gated recurrent units (GRUs [34]) in several benchmark tasks. Furthermore, in [35], different plasticity rules and learning rates are evolved within the same reservoir, thus allowing flexibility in the type and rate of neural plasticity. All of these approaches for task adaptive RC are software simulated. Although software simulated reservoirs offer enough flexibility to implement such task-dependent adaptive rules, doing the same in physical RC poses a significant challenge. However, recently adaptive reservoirs have also been implemented in physical systems by [36]. By adapting the thermodynamical phase space of the physical substrate, the same reservoir could be optimized for multiple tasks.

More generally, Hebbian plasticity is based on local learning rules: a weight update strategy inspired by biology that only make use of local information (i.e., information present at the synapse). This is generally understood to be pre- and postsynaptic firing or firing frequency. Since there is usually no error signal, local learning rules are suitable for unsupervised learning. The combination of adaptive reservoirs based on unsupervised local learning rules with supervised readout layers combines the benefits of both learning schemes, while still avoiding the computational cost of backpropagating through the entire reservoir. Adaptive DDNs (ADDNs), introduced in [22], have adaptive reservoirs with delay-sensitive BCM synapses in addition to delay connections, allowing the reservoir to adapt to input dynamics. Delay-sensitive BCM differs from the regular BCM rule (see [32]) in what is considered local information (see Figure 1). Since delays (modeled by physical distance) between the pre- and postsynaptic neuron are included, the presynaptic rate is not local information at the synapse. Instead, the presynaptic activity delayed by *d* timesteps is used, where *d* is the amount of delay applied by the connection, resulting in the following weight update equation.
(5)$$\begin{array}{cc} \Delta W_{D=d} & =\eta ⊙x\left(n\right)⊙\left(x\left(n\right)-\theta_{M}\left(n\right)\right)x^{T}\left(n-d\right) \end{array}$$

Here, $\Delta W_{D=d}$ is the weight change for all weights with *d* delay steps (i.e., all elements corresponding to connections of different delays are set to 0), ⊙ is the element-wise multiplication operator, η is the matrix of learning rates (i.e., each weight is updated according to a separate learning rate), and $\theta_{M}\left(n\right)$ is the adaptive threshold at time *n*, defined in Equation (Equation 6).
(6)$$\begin{array}{cc} \theta_{M}\left(n\right) & =E^{2}\left[\left(x\left(n\right)/y_{o}\right)\right]\approx {\left(\frac{1}{T}\sum_{t=0}^{T}\frac{x(n-t)}{y_{o}}\right)}^{2} \end{array}$$
where $y_{0}$ is a scaling parameter, and E[·] is the expected value. This is estimated using a sliding average with a time window of *T* simulation steps.

The software implementation of BCM in DDNs is provided in the Appendix A.

### 2.3. Training and Optimization

DDN readout layers are trained using the same procedure as in conventional ESNs. Task input is fed into the reservoir, with the first 400 reservoir states being discarded to account for the transient dynamics. A ridge regression model is fitted to predict the task labels based on the reservoir activity. Next, the network activity is reset, and the readout task performance is evaluated on a validation set, with the first 400 reservoir states again being discarded.

As mentioned, this procedure is performed for each hyperparameter candidate evaluation, such that each candidate can be assigned a fitness score based on a specified validation performance metric. Using CMA-ES, the hyperparameters are optimized to maximize this fitness score. The hyperparameters that need to be optimized for DDNs consist of the standard ESN hyperparameters (e.g., weight scaling and connectivity), as well as location-specific hyperparameters (e.g., location mean and variance). However, DDNs are based on a multi-reservoir approach. In this case, the hyperparameters are not defined for the entire reservoir, but the reservoir consists of several sub-reservoirs, or clusters of neurons. As such, instead of having for example, a single connectivity parameter, the connectivity is defined between and within each sub-reservoir. The location of the neurons can then be sampled from a Gaussian Mixture Model (GMM), with each of the GMM components corresponding to a sub-reservoir. This introduces the mixture weights as additional hyperparameters. A detailed overview of DDN hyperparameters as a function of number of sub-reservoirs *K* is given in Table 1.

Next to readout training (supervised), ADDNs require reservoir adaptation (unsupervised). As such, the training procedure for each reservoir consists of an unsupervised pre-training step where task data are fed into the reservoir, without recording reservoir states. Reservoir weights are updated according to delay-sensitive BCM. After pre-training, reservoir weights are fixed, and the readout layer is trained as before.

In addition to the DDN hyperparameters presented in Table 1, ADDNs include learning-rate hyperparameters, which indicate how adaptive each cluster to cluster projection is, and an adaptive threshold scaling parameter, defined for each cluster. These are optimized together with all the previously introduced hyperparameters, similar to the approach presented in [35].

### 2.4. Tasks

In this paper, we use the following benchmark tasks to quantify and analyse reservoir performance.

Mackey–Glass Signal Generation is a commonly used benchmark task for RC. The Mackey–Glass timeseries are generated using the following Equation [37]:(7)$$\frac{dx}{dt}=\beta \frac{x(t-\tau )}{1+x{\left(t-\tau \right)}^{n}}-\gamma x\left(t\right)$$
where the parameters were τ=17, n=10, β=0.2 and γ=0.1. However, because we simulate in discrete time, we make use of the discrete version of this equation, with τ being a positive integer.
(8)$$x\left(t+1\right)=x\left(t\right)+\beta \frac{x(t-\tau )}{1+x{\left(t-\tau \right)}^{n}}-\gamma x\left(t\right)$$

For the signal generation task, the goal is to generate Mackey–Glass sequences as far into the future as possible, based on a limited number of previous steps, without receiving new feedback during signal generation. This is achieved by first training the network readout layers for one step ahead prediction. During validation, new Mackey–Glass state estimations are re-used as inputs for subsequent predictions. We refer to this as ‘blind prediction’. We evaluate task performance using the Prediction Horizon metric introduced in [22], which is the amount of blind prediction steps that the network can perform, while maintaining an absolute prediction error lower than a pre-defined error margin. In this paper, we use an error margin of $0.1\sigma_{l}^{2}$, where $\sigma_{l}^{2}$ is the label variance.

NARMA system approximation is another commonly used RC benchmark task. NARMA stands for nonlinear auto-regressive moving average and is characterized by the order, which defines the temporal dependencies in the system. In this paper, we use 10th order NARMA (NARMA-10) and 30th order NARMA (NARMA-30) systems, which are defined by, respectively,
(9)$$\begin{array}{cc} y(t+1) & =0.3y\left(t\right)+0.05y\left(t\right)\sum_{i=0}^{9}y\left(t-i\right)+1.5u\left(t-9\right)u\left(t\right)+0.1 \end{array}$$
(10)$$\begin{array}{cc} y(t+1) & =0.2y\left(t\right)+0.04y\left(t\right)\sum_{i=0}^{29}y\left(t-i\right)+1.5u\left(t-29\right)u\left(t\right)+0.001 \end{array}$$
where y(t) is the system state at time *t*, and u(t) is the system input at time *t*, which is uniformly distributed between 0 and 0.5. The goal in this task is to predict the next state of the NARMA system based on the previous, serially presented inputs. During evolution, the reservoirs are trained and validated using the same input as given to the NARMA systems.

### 2.5. Linear Memory Capacity

One of the metrics commonly used to quantify reservoir dynamics independent of tasks or read-out layers is memory capacity (MC) [23,26]. We use Equation (Equation 11) for total linear MC and Equation (12) for lag-specific MC.
(11)$$\begin{array}{cc} MC_{tot} & =\sum_{k=1}^{\infty }MC_{k} \end{array}$$
(12)$$\begin{array}{cc} MC_{k} & =r{\left(u\left(n-k\right),{\hat{u}}_{k}\left(n\right)\right)}^{2} \end{array}$$

Here, r(·,·) is the Pearson correlation coefficient, u(n) is the input to the reservoir at time *n*, and ${\hat{u}}_{k}\left(n\right)$ is the output generated by a readout layer trained to reproduce the input from *k* steps ago, or u(n−k). We refer to the term *k* as lag, which is simply the time difference between the current input, and the past input to be reproduced. Because the readout layer consists of a simple linear regression for each reservoir, the MC is considered to be a property of the reservoir, independent of the readout layer [26]. The input to be reproduced that is used for computing MC should be independently and identically distributed (i.i.d.).

In theory, total linear MC is defined as the sum of memory capacities for lags 1 to *∞*. This sum is upper bounded by the number of observed reservoir states. In a conventional ESN, this equals the number of neurons. For the experiments presented in this paper, we only report on MC values up to a fixed maximum lag of 150. This corresponds to $6D_{\max}$.

Memory capacity is a measure of how much short-term memory a reservoir has, which is defined by how well the input at time n−k can be reproduced based on reservoir activity at time *n*. Two systems with the same total MC can have different capacity profiles (i.e., different values for $MC_{k}$ for each *k*) and can be useful in different types of tasks, as tasks can require linear memory from different points in the past. In regular ESNs, it has been shown that the total linear MC of a reservoir is maximized when all reservoir nodes are linear and serially connected [28]. More generally, linear MC is in trade-off with non-linear computation [26]. Simply maximizing total linear MC is not useful, as real-world tasks generally require not just recalling as many past inputs as possible but recombining inputs from specific times, often also with non-linear transformations. Without non-linear transformations, reservoirs are not able to increase the linear separability of the input data. It has been shown that mixture reservoirs, which combine non-linear and linear nodes, considerably outperform purely non-linear reservoirs [38].

With the same number of non-linear nodes in a reservoir, the amount of information and duration for which any information is kept in the system unchanged (i.e., transformed only linearly) is greatly increased by adding inter-node delays. In fact, the main function of biological axons is to propagate signals through space without performing any integration or non-linear computations. However, axons can be found with vastly different lengths and propagation speeds depending on several properties such as diameter and insulation. Similar to this phenomenon, one of the proposed effects of distributed distance-based delays in reservoirs is the increase in MC without the reduction of non-linear computation, since the addition of delays does not reduce the number of non-linear nodes in the reservoir.

## 3. Results

In this section, we first describe the experimental setup used to obtain the optimized networks. Based on these networks, we show that the addition of delay increases total linear memory capacity. Secondly, we show that delays evolve such that reservoir’s MC profile starts to match the tasks memory requirements. To this end, we computed the linear MC profiles of the DDN models that we optimized on the NARMA system approximation task and the (A)DDN models optimized for the Mackey–Glass signal generation task. To quantify the memory requirements of a task, we define the task capacity profile. This is simply the correlation between the correct task output y(n) (i.e., label) and the delayed input u(n−k), delayed by some lag *k*, plotted against *k*.
(13)$$\begin{array}{cc} TC_{k} & =r{\left(u\left(n-k\right),y\left(n\right)\right)}^{2} \end{array}$$

In case of auto-prediction tasks, where the system state is the input at the next time point (i.e., u(n)=y(n−1)), this is equivalent to the autocorrelation function of the system state. For tasks where there is a separate input signal, this corresponds to the cross-correlation between the input and system state. Both of these functions are commonly used in reservoir computing as characterization of task memory requirements [18]. Since ESN readout layers are linear models, it is reasonable to assume that this input–output correlation profile of the task system indicates at which lags the most information about the next time point can be obtained.

### 3.1. Experimental Setup

We optimized the hyperparameters of a DDN, ADDN and conventional ESN on the Mackey–Glass signal generation task using CMA-ES, as described in Section 2.3. We used reservoirs consisting of 300 neurons and 4 clusters. CMA-ES populations consisted of 25 candidates and were evolved for 200 generations. Mackey–Glass training sequences for DDNs and ESNs consisted of 1500 samples, and validations sequences were 1000 samples. For ADDNs, we used pre-training sequences of 500 samples, readout training sequences of 1000 samples, and 1000 sample validation sequences. As such, the total amount of training samples is equal between adaptive and non-adaptive reservoirs. For each evaluation step (pre-training, training and validation), five sequences were used, and each sequence was preceded by 400 ‘warmup’ samples to account for initial transient dynamics. The validation performance of this evolution (in terms of prediction horizon) can be seen in Figure 2A. Here, we see results consistent with previous experiments, with ADDNs performing best (i.e., they achieve the highest prediction horizon), followed by DDNs and standard ESNs. Next, we optimized the hyperparameters of 50-node DDNs and ESNs for NARMA-10 and 100-node DDNs and ESNs for NARMA-30. Note that we do not optimize ADDNs for the NARMA tasks. Since NARMA systems use random, uniformly distributed input, there are no temporal patterns to be learned in the input data using unsupervised learning rules. Again, 25 CMA-ES candidates were generated per generation, and the reservoirs were evolved for 200 generations. A training sequence of 8000 samples and a validation sequence of 4000 samples were used in both experiments, again preceded by 400 warmup samples. The resulting validation NRMSE throughout CMA-ES evolution is shown in Figure 2B. Unless otherwise specified, an optimized model refers to a reservoir generated using the best performing candidate hyperparameter set.

### 3.2. Memory Capacity and Delay

The simplest way to demonstrate the effect of delay on linear MC is by observing what happens if delay is removed from a DDN optimized for Mackey–Glass signal generation. We measure $MC_{k}$ of an optimized DDN for k=[0,150], and we do the same for a delay-less, otherwise identical copy of this network. This network is obtained by generating a conventional ESN with the same reservoir, input and bias weights, and the same leak parameters as the DDN. We show their respective MC profiles in Figure 3. The total MC is lower in the network without delays. Moreover, we note that the capacity profile of the optimized DDN is concentrated at higher lags (with a prominent peak around 20 timesteps lag), compared to the delay-less network. The linear MC in conventional ESNs is (mostly) monotonically decreasing with lag, with the highest MC at a lag of 0, due to the fading memory property. Intuitively, if the relevant input was given further in the past, it has been altered by more non-linear activation steps and is thus harder to reproduce. This is not the case for DDNs and ADDNs. Due to inter-neuron delays, the MC profile can be non-monotonically decreasing with lag, instead having peaks and valleys in the MC profile, which we observe in Figure 3.

In Figure 4, we present the inverse of this experiment, adding random delays to an optimized ESN in two different network topologies. We take the ESN optimized for Mackey–Glass signal generation, and we create four DDN copies of this network. Recall from Section 2.1 that when generating DDNs, neuron locations need to be specified, as delays are calculated based on neuron locations. In the experiment shown in Figure 4A, the neuron locations were sampled from a 2D Gaussian with mean (0, 0), an x-axis and y-axis variance of the distance equivalent to one delay step, and 0 x-y correlation. The input neuron location is at (0, 0). A representation of the network layout can be seen in Figure 4B. For each of the four DDN copies, we multiply the same original neuron coordinates by, respectively, 25, 50, 75, and 100, thus maintaining their topology. This simply means that all the delays are scaled by the same factor. In Figure 4C, we repeated this experiment using an input neuron that is positioned far away from the reservoir, as shown in Figure 4D. We compare the memory capacity profiles of the nine resulting networks. Note that the DDN copies of the optimized ESN are not re-optimized with the added delays (i.e., the delays are not tuned to the task). Nevertheless, we see that this random introduction of delays increases the total linear memory capacity of the system, with more delay generally leading to higher memory capacity in both network types. With the topology shown in Figure 4B, we see that the MC profiles are simply stretched out over more lag values, while still having their first peak close to a lag of zero. In contrast, the topology from Figure 4D shows a smaller increase in total linear MC, but the MC profiles are shifted towards higher lag values, such that the first peak occurs later. With the distal input neuron, most of the delay is present at the input weights, such that inputs simply arrive later in the reservoir, thus delaying any reservoir response useful for input reproduction. Note that for these experiments, we do not include any performance measures on the task that these networks were originally optimized for (in this case Mackey–Glass). Altering the optimized networks by randomly introducing or removing delays means that they need to be re-optimized for the task. As such, here, we are not interested in the task-performance.

In conclusion, we see that removing delays from an optimized DDN decreases (linear) memory capacity and that adding delays to an optimized ESN increases it. Moreover, we observe a shift in the memory capacity profile towards larger lags as we increase the distances between neurons. In addition, the linear MC profiles become increasingly spread out over larger lag ranges with increasing inter-neuron distance. In contrast, the linear MC profile of the task optimized DDNs presented in Figure 3 show sharp peaks of concentrated MC instead of spread-out profiles. This suggests specific task memory requirements at the respective lags. As such, in the following section, we analyze the relation between task memory requirements and linear MC.

### 3.3. Memory Capacity and Task Capacity

As mentioned in Section 1, we suggest that variable-delay-based models evolve such that their MC profile matches the task memory requirements. To test this hypothesis, we computed the linear memory capacity profiles of ESNs, DDNs and ADDNs throughout evolution, for the best model of every tenth generation. During ADDN evolution, before training the readout-layer, each candidate ADDN reservoir is adapted to input dynamics using delay-senesitive BCM in an unsupervised pre-training stage. Similarly, ADDN MC profiles are computed for adapted networks (i.e., after using delay-sensitive BCM).

In all three cases (ESNs, ADDN and DDN), for each hyperparameter candidate solution, the MC profiles are computed and averaged over five reservoirs. The MC profiles are computed up to a lag of 100. In the following subsections, we first discuss the results for the NARMA tasks. These tasks make use of uncorrelated input sequences, and as such, the results are clearly interpretable. Next, we discuss the results for the Mackey–Glass signal generation task, for which the networks use self-generated predictions as new inputs. Because these networks are optimized for such temporal dependencies in the input data, the resulting MC profiles are harder to interpret.

#### 3.3.1. NARMA

As mentioned in Section 3.1, we made use of 100-node DDNs and baseline ESNs for NARMA-30 and 50-node DDNs and baseline ESNs for NARMA-10. We decreased the reservoir size for NARMA-10 so that the memory requirements of the task are high enough, since the number of nodes has a large influence on total memory capacity. With enough nodes, the reservoir will be able to evolve towards a memory capacity of 1 up until the largest lag necessary. This obscures any effect that the evolvable inter-node delays have on matching memory capacity to task capacity. We evaluate the linear memory capacity of the best candidate in every tenth generation. The results for both DDNs and ESNs evolved for NARMA are shown in Figure 5A–D. The task profile is shown in Figure 5E,F. For conventional ESNs (Figure 5C,D), we see monotonically decreasing memory capacities throughout evolution. For ESNs evolved for NARMA-30, we see that the lag of 30, which corresponds to the second peak in task capacity, is reached in later generations. In the case of DDNs, however, we see a clear alignment between the evolved memory capacity profiles and the task profiles, for both tasks. For the first few generations, the not yet evolved networks show a more ESN-like MC profile without distinct peaks, which start to shift to higher lags and finally splits into two peaks, coinciding with the peaks in the task profile. We see a third intermediary peak around lag 15 in the case of the NARMA-30 evolution. This can be explained by the fact that not enough delay was present in the reservoir to cover the full gap from lag 6 to 30. The maximum possible delay in this setup was 25, which means that the reservoirs had to evolve to reach a total lag of 30 with two sets of delayed connections.

#### 3.3.2. Mackey–Glass

For the DDNs, ADDNs and ESNs optimized for Mackey–Glass, the results are shown in Figure 6A, Figure 6B and Figure 6C, respectively, represented as a heatmap with generation on the y-axis and lag on the x-axis. In Figure 6D, we show the task capacity profile of the Mackey–Glass sequences used during hyperparameter optimization. Note that because the Mackey–Glass input at time *n* is the output at time n−1, applying Equation (Equation 13) on the Mackey–Glass system is equivalent to an autocorrelation profile, as the input from *k* steps ago u(n−k) is equivalent to the output from k+1 steps ago y(n−k−1). Again, we expect to see that throughout evolution, the linear MC profiles start to overlap with the task capacity profile.

Similar to the NARMA experiments, we observe monotonically decreasing linear MC profiles for the baseline ESNs (Figure 6C). In Figure 6A,B, we observe that the (A)DDNs start out with a MC profile similar to that of a regular ESN: highest for a low lag and monotically decreasing. In later generations, peaks in MC form that start shifting towards the lags corresponding to peaks in task capacity, where ADDNs show slightly better overlap with the first task capacity peak. Additionally, we see that ADDNs evolve towards a lower total linear MC compared to DDNs. This could be due to more non-linearity present in the reservoirs, which could lower linear MC.

In this experiment, the final MC profile and task profile only somewhat overlap. One possible explanation is that there is not enough delay present in the system to achieve such a large gap between MC peaks. Recall that a non-monotonically decreasing MC profile is not possible with conventional ESNs and is only made possible here due to inter-node delay. The maximum distance between two peaks in MC depends on the maximum delay present in the reservoir.

Furthermore, note that the networks are trained to carry out one-step ahead prediction (i.e., readout parameters are chosen to minimize one-step-ahead prediction error), but are validated using the measure of prediction horizon (i.e., the network hyperparameters are optimized to maximize the prediction horizon). However, prediction horizon is defined as the amount of blind prediction steps the network can perform, while maintaining a low enough error. Although one-step-ahead prediction can be carried out based on the past actual inputs, blind prediction must be carried out based on past predicted inputs, since past predictions are used as new inputs. The past inputs that the reservoir received are not inputs generated by a perfect Mackey–Glass system, but self-generated inputs of the current reservoir. The currently used Mackey–Glass task capacity shows for which lags the past labels of this task give most information about future states. As such, it might not be a good approximation of the true task capacity. A better approximation would be the correlation between the Mackey–Glass labels (i.e., the actual sequence) and the past blind predictions generated by the DDNs (i.e., the inputs used). This shows how well we could predict the next label based on each of the past inputs. However, since the past DDN predictions are not perfect, and the errors are fed back into the input, we expect to see the error compound over time. Due to this compounding error, the correlation between past predicted Mackey–Glass states and real Mackey–Glass states can differ from the Mackey–Glass autocorrelation shown in Figure 6. However, the amount of error that compounds during blind prediction differs per model and decreases throughout evolution. Moreover, it is known that the use of non-i.i.d. inputs complicates the characterization of the memory requirements of tasks [39].

In conclusion, due to the nature of non-i.i.d. input in a signal generation task, it is hard to disentangle the causes for the slight mismatch of task capacity profiles and memory capacity profiles. In contrast, the experiments presented in Section 3.3.1 made use of i.i.d. inputs instead of using the next state as inputs. This resulted in a very clear overlap between task capacity and memory capacity.

## 4. Discussion

In this work, we first reproduced earlier findings on DDNs and ADDNs. We showed that DDNs outperform conventional ESNs on the Mackey–Glass blind prediction task, and that ADDNs outperformed DDNs. Since these models are stochastic in nature and require long hyperparameter optimizations, any replication of the findings presented in [21,22] is useful, showing that these results are consistent.

Secondly, we investigated the earlier presented hypothesis that DDNs, through the use of delays, can overcome the limitations associated with the memory–nonlinearity trade-off, by increasing memory capacity without increasing the number of nodes or switching non-linear nodes with linear nodes. To test this, we showed that total linear MC is lowered by removing the delays from an optimized DDN. Next, we randomly added variable inter-node delays to an optimized conventional ESN and showed that total linear MC grows as we increase the delays. This shows the effect of delay on linear MC, even without any optimization. Moreover, in both cases, we see that delays allow for non-monotonically decreasing memory capacity profiles. With delays in input connections, as seen in Figure 4C,D, we observe MC profiles that are shifted towards larger lags. However, specifically inter-neuron delays inside the reservoirs allow for multiple peaks in memory capacity (see Figure 4A,B), whereas input delay only shifts the first peak towards larger lags, while keeping the rest of the MC profile monotonically decreasing. Herein lies an essential difference between DDNs and previously explored delay reservoirs, such as discussed in [18,20]. As such, we see that input delays facilitate tasks that require memory at larger lags but fail to provide the reservoir with memory at lags that are farther apart.

Next, we proposed that DDNs and ADDNs optimized with CMA-ES evolve such that the memory capacity profiles of their reservoirs match the task profile (i.e., the memory requirements of the task, represented as correlation between the task output and lagged input, plotted against lag). We showed that the peaks in DDN and ADDN memory capacity profiles indeed start to move towards peaks in the task capacity profile as they evolve. In the case of DDNs optimized for the NARMA system approximation tasks, where we have random, uncorrelated inputs, we see that the linear MC profiles quickly match their task capacity profiles after several (approximately 40) generations of CMA-ES hyperparameter optimization. This supports the hypothesis that delays can be evolved to increase the memory of reservoirs, in a way that matches the memory requirements of tasks. However, in the case of (A)DDNs optimized for Mackey–Glass signal generation, we do not observe a clear overlap. This is likely due to the non-i.i.d. nature of the input sequences, and the compounding prediction error associated with blind prediction tasks.

The tuning of intra-reservoir delays to specific task requirements could be exploited in physical reservoir computing applications. All physical systems inherently implement some delay. Consider that the delays investigated in this paper do not each require individual tuning. Instead, for each cluster to cluster projection, they are sampled from a distribution. Similar to other reservoir hyperparameters, the tuning of individual delays is tolerant to a high degree of stochasticity. Optimizing the within- and between-cluster timescales for specific tasks should be achievable prior to implementation in physical systems and could help to reduce the cost of finding task-optimal physical reservoirs. Several physical reservoir realisations, including all-optical [40] and electrical [16], have shown the benefits of delay tuning for input or feedback delays.

Future work should study how delays affect non-linear processing capacities of reservoirs next to the effects on memory by using non-linear memory capacity measures. Next, it is necessary to further analyze the effect of reservoir adaptation in ADDNs on memory capacity, e.g., by studying the change in MC profiles as ADDNs adapt to changing input dynamics. Lastly, as mentioned in [39], autocorrelation in the task system makes it difficult to characterize task memory requirements. Therefore, it is important to further investigate how the temporal properties of tasks relate to optimal delay and memory capacity in reservoirs.

## Figures and Tables

**Figure 1 biomimetics-09-00355-f001:**
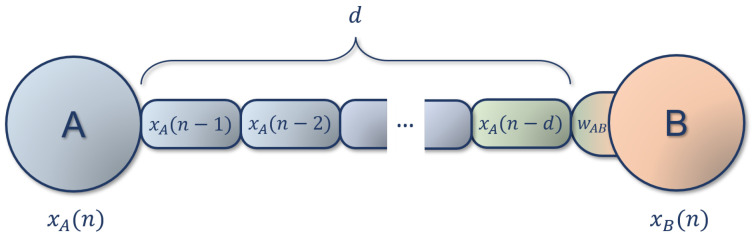
Schematic representation of a(n) (A)DDN connection, with a delay of *d* time steps. The activity of neuron A is present at the input of neuron B after *d* steps, so the input for neuron B at time *n* will be $x_{A}\left(n-d\right)$. In the case of ADDNs, where connections can be adaptive, the weight change $\Delta w_{AB}$ is computed based on the postsynaptic activity $x_{B}\left(n\right)$ (represented in orange), and the delayed presynaptic activity $x_{A}\left(n-d\right)$ (represented in green).

**Figure 2 biomimetics-09-00355-f002:**
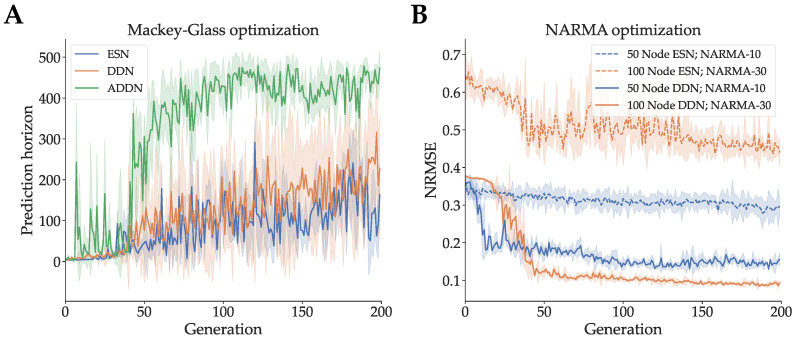
Validation performance throughout CMA-ES hyperparameter optimization (evolution). The x-axes refer to the CMA-ES generations. The shaded areas represent the standard deviation in performance of the best candidate of the generation. (**A**): DDN, ADDN, and baseline ESN Mackey–Glass Performance expressed in prediction horizon (e.g., number of blind prediction steps during validation until error margin is reached). These reservoirs are evolved to optimize task performance on the Mackey–Glass signal generation task, where the goal is to predict as many future states of the Mackey–Glass system (Equation (Equation 8)) as possible based on previous states, while maintaining a low enough absolute error. (**B**): DDN and baseline ESN performance on NARMA system approximation tasks expressed in normalized root mean squared error (NRMSE). The goal is to predict the next output of the NARMA system (Equations (Equation 9) and (10)), based on all previous inputs. We use 50 node and 100 node ESNs and DDNs for the NARMA-10 and NARMA-30 task, respectively.

**Figure 3 biomimetics-09-00355-f003:**
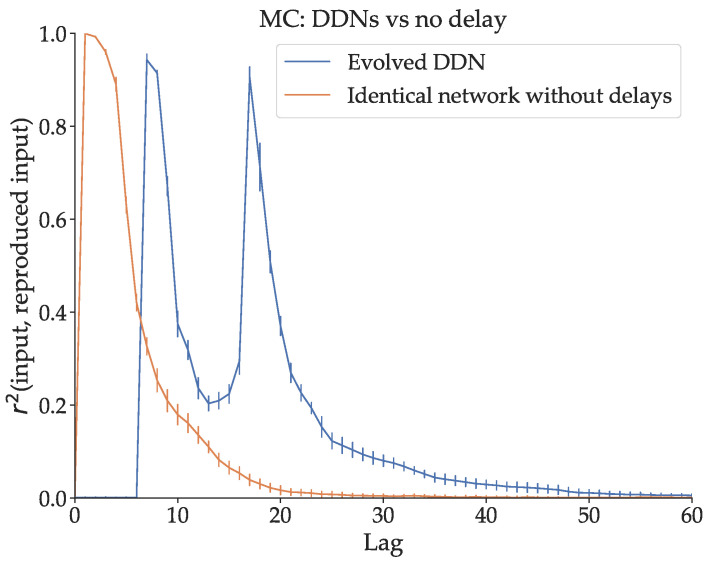
A comparison between the memory capacity profile of a DDN optimized for Mackey–Glass with a maximum delay of 25, and a conventional ESN with the same parameters (i.e., a non-delayed copy). The linear memory capacity is averaged over 20 trials, with the error bars representing standard deviation. Here, the average total memory capacity ($MC_{tot}$, the integral of the MC profile, averaged over the 20 trials) of the DDN and the ESN is, respectively, 18.92 and 11.82. We see that the optimized DDN has a concentration of higher memory capacity at higher lags and is spread out over more lags. In contrast, the memory capacity of the delay-less network is concentrated on smaller lags and spans fewer lag values.

**Figure 4 biomimetics-09-00355-f004:**
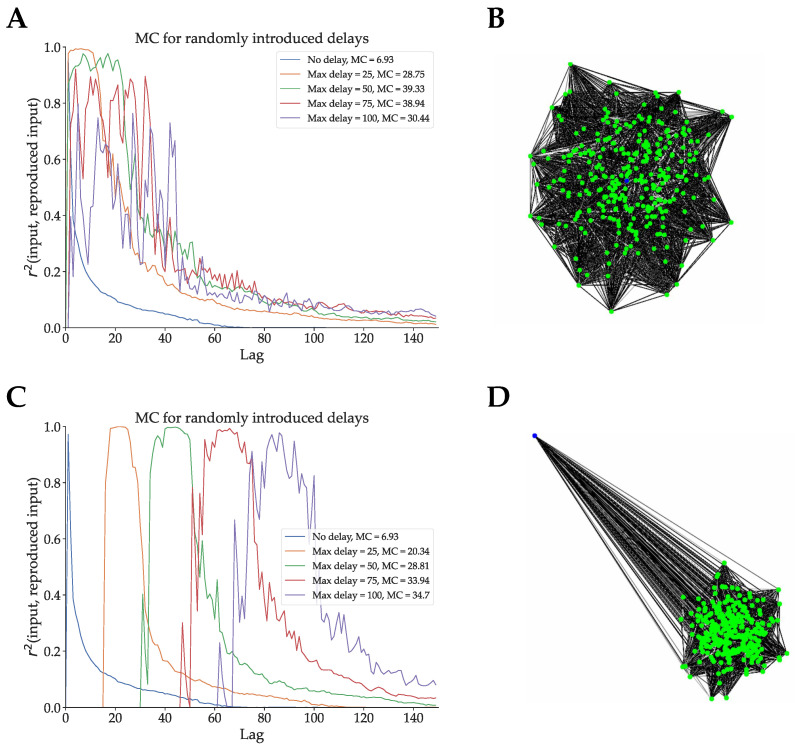
We optimized conventional ESNs for 200 generations with CMA-ES. The best candidate hyperparameter set was used to generate a network. (**A**): Four DDN copies with normally distributed neuron locations and different (spatial) network sizes were made, with the input neuron in the centre of the network. In this graph, we show the linear MC profiles of these original ESN and the four DDNs of increasing spatial dimensions. The total linear MC is mentioned in the legend. (**B**): A spatial representation of the network corresponding to (**A**). We see that the input neuron (blue) is located in the centre of the normally distributed reservoir neurons (green). (**C**): Analogous to (**A**), this graph shows the linear MC profile of increasingly larger DDNs. In this case, however, the input neuron (blue) is in a distant position. (**D**): A spatial representation of the network corresponding to (**C**).

**Figure 5 biomimetics-09-00355-f005:**
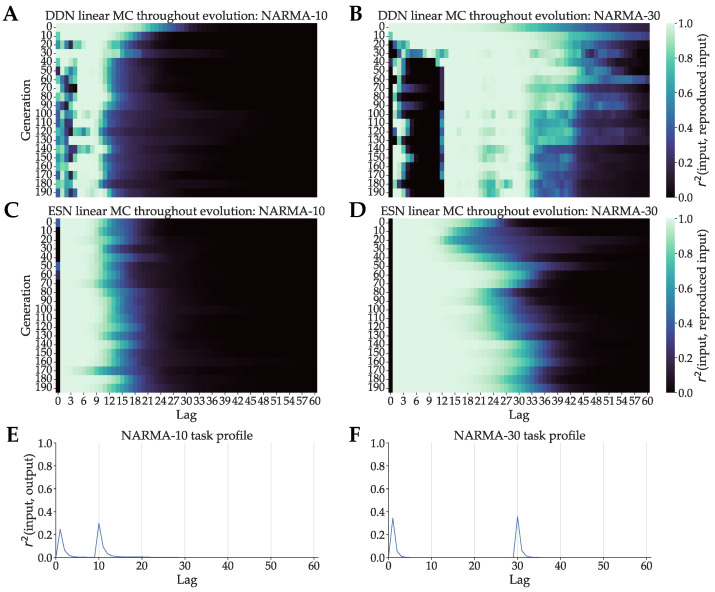
DDN linear memory capacity profiles throughout evolution optimized for NARMA system approximation task. In (**A**,**B**), we see the DDN linear MC profiles of the best performing candidate hyperparameter set of every tenth generation. These were computed based on five networks generated from each candidate. MC was computed up to a lag of 100 and averaged over the five networks. (**C**,**D**) show the same for baseline ESNs for the same tasks. (**E**): the NARMA-10 task profile, computed as the squared correlation between lagged input and output, plotted against lag. (**F**): Analogous for NARMA-30 task.

**Figure 6 biomimetics-09-00355-f006:**
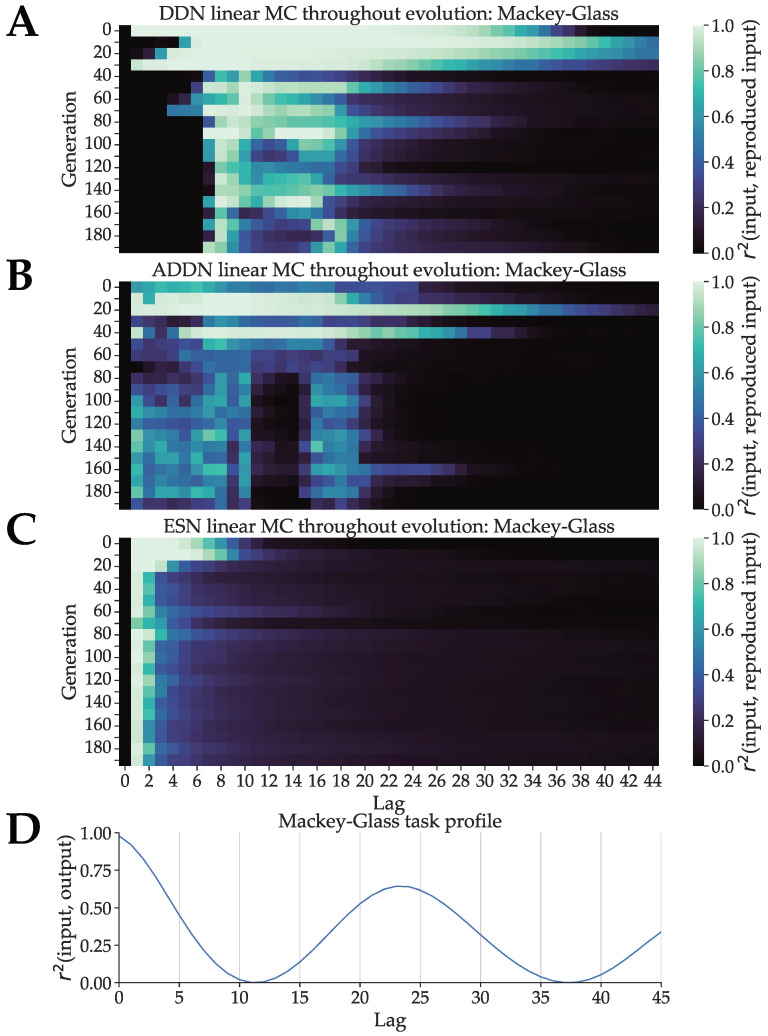
(**A**): DDN linear memory capacity profiles throughout evolution, optimized for the Mackey–Glass signal generation task. The best performing candidate hyperparameter set of every 10th generation is used to generate five networks. The linear memory capacity profiles up to a lag of 100 is computed for each of these five networks and averaged. (**B**): Analogous for ADDNs. In this case, reservoirs are first adapted using delay-sensitive BCM before computing MC. (**C**): Analogous for baseline ESNs. (**D**): The task capacity profile of the Mackey–Glass system. This represents the correlation between the lagged state of the system with the current state of the system.

**Table 1 biomimetics-09-00355-t001:** DDN hyperparameters as a function of number of reservoir clusters *K*.

Standard ESN Hyperparameters		
**Name**	**Shape**	**Description**
Weight scaling	*K* by *K*	Weight scaling factor for weights from cluster *i* to cluster *j*.
Bias scaling	*K*	Bias scaling factor for each cluster.
Connectivity	*K* by *K*	Fraction of non-zero weights for connections from cluster *i* to cluster *j*.
Decay	*K*	Decay/leak parameters for each cluster.
**Location-Related Hyperparameters**		
**Name**	**Shape**	**Description**
Component means	*K* by 2	Location mean of each cluster.
Component variance	*K* by 2	GMM component variance along the *x*- and *y*- axis.
Component correlation	*K*	GMM component *x*- and *y*-axis correlation.
Mixture weights	*K*	Mixture weights define how neurons are distributed over GMM components.

## Data Availability

The raw data supporting the conclusions of this article will be made available by the authors on request.

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
