# Peer review of "Exploiting Signal Propagation Delays to Match Task Memory Requirements in Reservoir Computing"

_biomimetics, 2024, doi:10.3390/biomimetics9060355_

Round 1

Reviewer 1 Report

Comments and Suggestions for Authors

My comments you can find in the attached PDF file.

Comments on the Quality of English Language

There are some places which need attention. In the PDF file, I addressed two cases. 

Author Response

Dear reviewer,

Thank you for your comments. Our response can be found in the document attached.

Reviewer 2 Report

Comments and Suggestions for Authors

The paper written bei S. Iacob and J. Dambre presents an interesting method to improve the time series prediction performance of a ESN via adding propagation delays. Also, the memory requirement of different tasks are discussed and relate dto the delayed variables needed for improvment. The paper is well written and the results are very interesting to the reservoir computing community. I have some questions and comments that the authors should answer before I can recommend publication. 

-In case of an experiment where the ESN network itself cannot be changed, the method could be applied to only one input (e.g. the effect of a delayed input is investigated here: 10.1088/2634-4386/ad1d32 or 10.3389/fams.2024.1221051). Are similar results possible in that case?

-The task capacity introduced in Eq.(12) appears to be identical to the autocorrelation function (ACF) for a time series prediction task with one variable. Please comment on that. Since the ACF has been used in other works about reservoir computing in the past, especially in the context of Takens embedding theory, this should be mentioned and discussed.

-Fig4: The information about how much the performance is improved in the different cases should be added.

-As stated by the authors and also discussed in 10.1515/nanoph-2022-0415, a non-iid input makes it hard to derive the memory requirements from the task time series. The delays needed are a complex interaction between the task and the reservoir timescales. I wonder if there are more insights that can be gained from the delay distrubution that evolves after training, e.g. in Fig.6, about that question.

- Some figures look a bit preliminary and could be improved for the final publication. e.g. Fig2, Fig3. there are also error bars missing in the plots that allow to judge about the variations with different realizations.

Author Response

(The authors gave the same response as above.)

Reviewer 3 Report

Comments and Suggestions for Authors

The manuscript is devoted to empirical research of the distance-based delay networks (DDNs) properties. The topic corresponds to the aims and scope of the Biomimetics journal. However, the scientific level of research conducted does not allow us to recommend the manuscript for acceptance in its current form. Indeed, the authors do not propose any new methodology, but only check some assumptions regarding DDN architectures by empirically testing a number of hyperparameter configurations.

There are a few additional comments:

1. At the end of Introduction, it is necessary to add a description of the further structure of the manuscript.

2. The authors should more clearly present the contribution of this research to the subject area.

3. It is necessary to revise fully the structure of the sections of the manuscript. The manuscript should contain sections with statements of the research problem, the proposed methodology as well as NN architectures for solving it. Only then specific datasets and their processing with results obtained should be presented.

4. What is the authors' contribution to Section 2 (Background)? What is the novelty of the methodology?

5. Discussion should include issues about applications for the developed methods.

6. The manuscript does not compare the results with the known state-of-the-art NN architectures.

Author Response

(The authors gave the same response as above.)

Round 2

Reviewer 3 Report

Comments and Suggestions for Authors

The response about the problem statement is not acceptable. Please, use mathematical formulation. It is also not clear why a comparison with other methods has not been made, since the manuscript considers a modification of the ESNs. Some comments regarding this in the paper would be convenient.

Author Response

Dear reviewer,

Please find our response to your feedback in the file attached.

Kind regards,

Stefan Iacob

Round 3

Reviewer 3 Report

Comments and Suggestions for Authors

All responses are clear.